# Thermoacid Behavior of Serpentinite of the Zhitikarinsky Deposit (Kazakhstan)

**DOI:** 10.3390/molecules29163965

**Published:** 2024-08-22

**Authors:** Abdrazak Auyeshov, Kazhymuhan Arynov, Chaizada Yeskibayeva, Kurmanbek Alzhanov, Yerkebulan Raiymbekov

**Affiliations:** 1Scientific Research Laboratory “Applied Chemistry”, M. Auezov South Kazakhstan University, Shymkent 160012, Kazakhstaneplusr@bk.ru (Y.R.); 2“Institute of Innovative Research and Technology” LLP, Almaty 050010, Kazakhstan

**Keywords:** serpentinite, thermal acid treatment, periclase, forsterite, sulfuric acid

## Abstract

Thermoacid behavior of serpentinite from the Zhitikarinsky field (g. Zhitikara, Kazakhstan). The character of dissolution of heat-treated serpentinite in a narrow temperature range of 600–750 °C is investigated, where the crystal lattice of the structural structure of chrysotile in sulfuric acid is destroyed. The X-ray and chemical analysis of the products of dissolution of heat-treated serpentinite at 600 °C, 725 °C and 750 °C in sulfuric acid solution show that the reason for the increase in the reactivity of heat-treated serpentinite at 725 °C and 750 °C with respect to the acidic medium and the degree of magnesium extraction into sulfate solution is the formation of periclase (MgO) in the serpentinite composition after heat treatment of them within a temperature range of 600–750 °C. The results were discussed using data obtained by conducting a thermodynamic evaluation of probable reactions during the thermoacid treatment of serpentinite, phase compressions of heat-treated serpentinite at 600–750 °C, and after its acid treatment at 1.0 M H_2_SO_4_.

## 1. Introduction

The chrysotile asbestos plant JSC “Kostanay Minerals” (c. Zhitikara, Republic of Kazakhstan) annually processes about 5.0 million tons of ore, containing mainly minerals of the serpentinite group (chrysotile, lizard, and antigorite). From chrysotile ore, 5–8% is extracted into commercial chrysotile asbestos fiber (the yield of the useful component from the total mass of processed ore). Together with the commercial production of chrysotile asbestos, up to 20% (depending on the fractions) of gravel is produced for sale. The rest are sent to the dump.

Since the ore is enriched only by machining, the chemical quality and quantity of the original ore do not change much and contain, on average, about 40% magnesium oxide. The chemical composition after enrichment varies within the following limits: MgO—34.0–40.0; Fe/FeO—4.2–5.8; MnO—0.14–0.36; Cr_2_O_3_—0.28–0.45; NiO—0.19–0.43; CaO—0.8–1.7; Al_2_O_3_—0.7–0.9; SiO_2_—43.18–44.0 [1]. There are breeds where the tails of enrichment can contain breeds with the inclusion of gold [2]. Therefore, the development of effective technological schemes for the extraction of useful components from the tails of chrysotile asbestos enrichment is one of the urgent problems for this chrysotile asbestos mining and processing enterprise, both in terms of the rational use of natural raw materials and environmental protection, and in carrying out the diversification of the main production.

It is not difficult to notice that this enrichment tail (waste) is primarily high-magnesium, which can potentially be used as a man-made raw material for the production of magnesium and its important compounds.

An analysis of research and development in recent years [3,4,5,6,7,8,9,10,11] devoted to this topic shows that although there is still no widely used technology in the world that allows the use of serpentinite as a source of magnesium and magnesium-containing products, there is a growing number of research and development activities aimed at finding ways of using it for this purpose. It is also noteworthy that the research focused on the selection of process parameters (type of serpentinite, degree of extraction of magnesium, purification of productive extraction solutions, temperature, concentration of acids, degree of grinding, duration of reaction, etc.). However, more recently, there have been frequent studies of the effects of thermoactivation of serpentinite before acid leaching [12,13,14,15,16], leading to an increase in the reactivity of serpentinite as compared to the original non-thermotreated. It is noted that this fact is important in solving the problems of serpentinite processing using acid methods. In order to fully substantiate the possibility of using serpentinite as a source of magnesium and its compounds, in this work, similar studies of serpentinite from the Zhitikarinskoye field (Kazakhstan) were conducted. The urgency of the need for such research is also determined by the fact that there is no quality dolomite deposit in Kazakhstan.

In this regard, the thermo- and acid behavior of serpentinite in the Zhitikarinskoye field (Kazakhstan) was investigated, including thermal, X-ray and chemical methods of research and analysis of thermo- and acid treatment products, as well as the estimation of the quantitative dissolution of the main elements of serpentinite constituents in acid. The combined methods (physicochemical and chemical) allow to establish the phase composition of complex systems of natural and newly formed inorganic compounds as a result of the thermal and acid treatment of serpentinite and establish a correlation between the thermal and chemical properties of serpentinite. Furthermore, the establishment of treatment temperature intervals that lead to optimal activation or reactivity of serpentinite relative to acid-base interactions is important in determining the effectiveness of serpentinite processing technologies using acid methods. The criteria include the influence of the treatment temperature on the duration of the leaching process, the degree of extraction of magnesium and other present elements from the serpentine raw material composition into the solution, and the conduct of technological processes in general.

The aim of this work is to study the influence of the processing temperature in the range of 600–750 °C, where the destruction of the crystal lattice of the structural structure of serpentinite occurs, on its reactivity and the nature of dissolution in sulfuric acid.

It is expected that the results of the research will be useful to enterprises in the chrysotile industry of Kazakhstan and other countries in the search for efficient and rational ways of using serpentinite ores and waste from their processing.

## 2. Results

### 2.1. Thermal Behavior and Observation of the Nature of the Dissolution of Heat-Treated Serpentinites in Sulfuric Acid

In the study of the thermal behavior of serpentinite heat-treated at t = 105 °C from the Zhitikarinsky deposit (Figure 1), it was shown that in the temperature range of 200–1000 °C, three obvious changes are detected according to the derivatogram, which are identified by processes: (1) at 350–450 °C with a maximum of 400 °C—decomposition of brucite, Mg(OH)_2_ (almost always found in asbestos-bearing rocks); (2) at 600–725 °C with a maximum of 660 °C—destruction of serpentinite or chrysotile structure Mg_6_Si_4_O_10_(OH)_8_; and (3) at 800–850 °C, the maximum of the exothermic process is 810 °C, the formation of forsterite Mg_2_SiO_4_.

Since the destruction of the chrysotile structure takes place within the temperature range of 600–800 °C, in this work, serpentinite samples were selected to determine the influence of the serpentinite treatment temperature on the acid treatment results: No. 1—at 105 °C was taken as the original; No. 2—heat treated at 600 °C; No. 3—heat treated at 725 °C and No. 4—heat treated at 750 °C. It was assumed that in this narrow temperature range (600–750 °C), in the area of fracture of the crystal lattice of the Mg_6_Si_4_O_10_(OH)_8_ structure and the emergence of new mineral phases, the thermodecomposed mass and neoplasms are in an unstable state, in which the bond between the newly converted primary silicate constituents’ anions (SiO_4_^2−^, SiO_3_^2−^) and magnesium cations (Mg^2+^) is maximally weakened or ruined. Further, it was interesting how the transformed composition of heat-treated (at 600–750 °C) serpentinite, formed from an unequal state depending on the treatment temperature, would affect the nature and parameters of the solution, especially the release of magnesium into sulfate solution, when they were acidified. This parameter is important in determining the technological and economic evaluation of serpentinite processing technologies using acid methods.

For this purpose, the above heat-treated samples No. 1–No. 4 were subjected to sulfuric acid treatment. Acid treatment of all samples was carried out under the same conditions (10 g of the heat-treated sample was dissolved in 1.0 M H_2_SO_4_, V = 100 mL), as mentioned in the description of the method of acid treatment.

External observation showed that when the sulfuric acid sample No. 1 (105 °C) is added to the solution, a gray-brown suspension is formed, which is well filtered in 5–10 min, the filtrate has a light blue tint. When a sample of No. 2 (600 °C) is added to the sulfuric acid solution, the suspension temperature rises to 88 °C.

When a sample of No. 3 (725 °C) is added to the solution under the same conditions, the reaction is instantaneous, the suspension boils, the reaction medium temperature rises to 95 °C, but the suspension is filtered very slowly, about 2 h. In this case, the leachate turns orange-yellow. When heated, it turns into an orange colloid, which on the next day turns from orange-yellow into a dark red gel.

When a sample of No. 4 (750 °C) is added to a sulfuric acid solution, the suspension boils up rapidly, being more reactive to the acid than previous samples, and the temperature rises to 95 °C, but it is also poorly filtered in 30 min. At the same time, the leachate turns red-brown, and when heated, it turns into a dark red gel. Figure 2 shows how the color of the resulting solutions changes when serpentinite is dissolved in sulfuric acid, depending on the heat treatment temperature of the original serpentinite.

As can be seen from the description of the external observations of the processes of the dissolution of the serpentinite under investigation, which are thermally activated at the selected calcination temperatures under the same conditions, they have a different nature of dissolution in sulfuric acid.

The observed interaction of serpentinite samples treated at a temperature range of 600–750 °C in a sulfuric acid solution clearly showed that the increased reactivity of serpentinite with respect to the acidic medium depends on the processing temperature in this narrow range, which is likely caused by the conversion of the original serpentinite composition after its thermal treatment, which has a higher alkaline property than the original serpentinite.

### 2.2. X-ray and Chemical Analyses of the Products of the Dissolution of Serpentinite in Sulfuric Acid

Figure 3 shows a diffractogram of the initial serpentinite (No. 1, 105 °C). The presence of the main phases is noted.

The diffractogram of the original serpentinite clearly spells out the interplane distance (MPR) of chrysotile Mg_6_[Si_4_O_10_](OH)_8_ with values d/n = 7,380–4,619–3,661–2,487–2,141–1,530 Å, which is the main phase and MPR of a significant amount of brusite—Mg(OH)_2_ with d/n = 4,770–2,365–1,794 Å. The diffractogram also shows overlapping MPR peaks of a small amount of antigorite d/n = 7.30–3.63–2.52 Å, a small amount of magnetite Fe[Fe_2_O_3_] d/n = 2.99–2.54–2.00–1.71–1.61 Å, as well as very weak MPR pyrope peaks—Mg_3_Al_2_[SiO_4_]_3_ values d/n = 2.92–2.69–1.50 Å and almandine—Fe_3_Al_2_[SiO_4_]_3_ d/n = 2.90–2.60–1.51 Å. Thus, 6 phases are present on x-rays in the original No. 1 specimen.

From the No. 1 K diffractogram obtained after treatment of initial serpentinite (No. 1) with sulfuric acid (1.0 M H_2_SO_4_) at a temperature of 80 °C, out of six phases peaks (MPR) d/n = 4,770–2,365–1,794 Å brusite—Mg(OH)_2_, the intensity of peaks MPR of chrysotile d/n = 7,380–3,661–2,487–1,530 Å and antigorite d/n = 7.30–3.63–2.52 Å. The diffractogram spells MPR peaks of the five phases—chrysotile, antigorite, magnetite, pyrop and almandin (Figure 4).

The calcination of the original serpentinite at a temperature of 600 °C (sample No. 2) leads to the decomposition of one of the main phases—brusitis, which can be seen from the diffractogram by a sharp reduction in the intensity of the MPR bursitis peaks at values d/n = 4,770–2,365–1,794 Å (Figure 5). Furthermore, peaks appear in the interphasic distance of the new phase, namely periclase (MgO), with values d/n = 2.43–2.10–1.48 Å, which is the product of the decomposition of brucite.

The diffraction characteristics of chrysotile, antigorite and magnetite do not change, and the characteristic MPR peaks of all three phases are recorded without any change. In this sample, the interplanar distance peaks of the pyrop (Mg_3_Al_2_[SiO_4_]_3_) d/n = 2.92–2.69–1.50 Å and almandine (Fe_3_Al_2_[SiO_4_]_3_) d/n = 2.92–2.69–1.50 Å are more clearly drawn.

When the sample No. 2 (600 °C) is treated with a sulfuric acid solution (1.0 M H_2_SO_4)_ in the diffractogram No. 2 K, no significant change is observed; all phases are drawn out without change, but the intensity of the MPR periclase, chrysotile and antigorite is reduced (Figure 6). Judging by the No. 2 K diffractogram, it can be believed that when leaching a sample of calcified at 600 °C 1.0 M with sulfuric acid, slow or partial dissolution of periclase, chrysotile and antigorite occurs (Figure 6).

More significant and interesting changes in the phase composition are found in the diffractograms of a sample of No. 3 tempered at 725 °C and No. 3 K obtained after treatment with a solution of sulfuric acid. The rise in calcination temperature up to 725 °C is accompanied by severe destruction of the main phase of the mineral—chrysotile (Figure 7). On the diffractogram No. 3 of the intense peaks of the interplanar distance chrysotile d/n = 7,380 (10)–3,661 (10)–2,487(10)–1,530 (10) Å only one is recorded, with a small clove at values d/n = 7,380 Å. On the diffractogram, new phases appear—d-tridemite (SiO_2_) with MPR values d/n = 4,390–4,120–3,730 Å, forsterite d/n = 3,875–3,470–2,753–2,497–2,441 Å, magnetite d/n = 2,990–2,541–2,097 Å, diopsid d/n = 2.99–2.89–2.56–2.04 Å, periclase d/n = 2.431–2.108–1.480 Å and MPR of pyrop d/n = 2.92–2.69–1.50 Å and almandin (Fe_3_Al_2_[SiO_4_]_3_) d/n = 2.92–2.69–1.50 Å are also recorded. Thus, in the composition of sample No. 3 perforated at 725 °C, the diffraction characteristics of 8 phases appear, the main phases become forsterite and magnetite.

After the sulfuric acid (1.0 M H_2_SO_4_) treatment of this sample No. 3 (725 °C) on the diffractogram No. 3 K (Figure 8), the MPR diopside peaks disappear at d/n = 2.99–2.89–2.56–2.04 Å, the MPR peaks of forsterite d/n = 3.87–3.47–2.75–2.44 Å and periclase d/n = 2,431–2,108–1,480 Å. In this case, the marked peak MPR of chrysotile at values d/n = 7.38–3.66–2.48–1.53 Å, but less intensity is maintained, which may be due to the fact that at a temperature of 725 °C, the chrysotile structure does not completely break down. It should be noted that the MPR peaks of pyrop d/n = 2.92–2.69–1.50 Å and almandine d/n = 2.92–2.69–1.50 Å rise. Hard-to-detect traces of MPR d-tridymite peaks with d/n = 4.39–4.12–3.73 Å are detected as there are large humps with shaded teeth in the area, possibly due to partial morphosis of the dilution products. After the acid treatment of sample No. 3 (725 °C), the characteristic peaks of the MPR of chrysotile, forsterite, magnetite, pyrop and almandin are preserved.

The calcination of serpentinite at 750 °C (sample No. 4) is accompanied by the complete completion of the phase change started at 600 °C (Figure 9). In X-ray diffraction characteristics of chrysotile and brusitis disappear, characteristic phase peaks which were already formed at a temperature of 725 °C remain, with values of interplane distance—forsterite d/n = 3,875–3,470–2,753–2,497–2,441 Å, magnetite d/n = 2,990–2,541–1,710 Å, diopsid d/n = d/n = 2.99–2.89–2.56–2.04 Å, periclase d/n = 2.43–2.10–1.48 Å, d-tridemite d/n = 4.39–4.12–3.73 Å, pyrop d/n = 2.92–2.69–1.50 Å and almandin d/n = 2.92–2.69–1.50 Å. According to the diffractogram, the composition of sample No. 4 hardened at 750 °C from composition No. 3 (725 °C) differs by the absence of the phase of chrysotile. The main phases are forsterite and magnetite. Thus, 750 °C is the completion temperature of the serpentinite phase transformation and the temperature of complete destruction of the crystal lattice of the chrysotile structure.

The diffractogram of this sample (Figure 10) after acid treatment of No. 4 K (750 °C) from the sample No. 3 K—725 °C (Figure 8) is distinguished by the fact that in the diffractogram the characteristic peaks of chrysotile MPR and diopside do not appear. The MPR peaks of forsterite d/n = 3,875–3,470–2,753–2,497–2,441 Å and periclase d/n = 2,431–2,108–1,480 Å decrease significantly, and the pyropa d/n = 2,920–2,690–1,500 Å and almandine d/n = 2.92–2.69–1.50 Å increase substantially. Of the diffraction characteristics of d-tridemite with MPR d/n = 4.39–4.12–3.73–2.49–1.69–1.52 Å, only the d/n = 1.69–1.52 Å peaks wer, due to large humps and shading of the diphractoma teeth in the 5–3 Å region.

Thus, in the heat treatment of serpentinite at 750 °C, the serpentinite structure of Mg_6_Si_4_O_10_(OH)_8_ completely breaks down, and it can be assumed that its dehydrocyclization products at 750 °C are subjected to acid action. Of particular interest was the change in the composition of samples (No. 3 and No. 4), which were perforated at 725 °C and 750 °C, where there are significant changes in the composition of serpentinite, affecting the physical and chemical processes of their dissolution in sulfuric acid and the composition of products.

Table 1 shows the results of the distribution of magnesium in the dissolution products of the above samples (No. 1–No. 4) in 1.0 M H_2_SO_4_ at 80 °C.

The results of the distribution of magnesium in the solution products, i.e., the quantity extracted in the magnesium sulfate solution and the magnesium content in the insoluble precipitation, show that, when the heat-treated samples are dissolved, the magnesium extraction ratio has (Figure 11a) sufficiently high values (84–87.5%), i.e., in the range of active integrity violation of the structural structure of the Mg_6_Si_4_O_10_(OH)_8_ crystal lattice. Thermoactivation of serpentinite in the region of the temperature of formation of forsterite leads to a gradual reduction in the transition of magnesium into sulfate solution. It should be noted that the main impurity metals (Fe, Al, Ca) found in sulfate solution decrease significantly as the calcination temperature of serpentinite rises (Figure 11b–d).

## 3. Discussion

The fact that the magnesium yield has a maximum value for samples of 725 °C and 750 °C when dissolved in acid may be due to the formation of low-temperature periclase and forsterite, formed relatively at lower temperatures than their high-temperature modifications, formed at t ≥ 800 °C, which can change the kinetic character of the dissolution of serpentinite in sulfuric acid with the course of ion exchange reactions in accordance with the equations:MgO + 2H^+^ → Mg^2+^ + H_2_O (1)
Mg_2_SiO_4_ + 4H^+^ → H_4_SiO_4_ + 2Mg^2+^(2)

Thermodynamic evaluation of probable periclase-to-serpentinite thermodecomposed reactions (MgO) in a temperature range of 600–750 °C suggests that possibilities for such reactions (Table 2) may exist.

The behavior of thermally treated serpentinite up to 600 °C in the interaction with acid and the products of its dissolution undoubtedly testifies that before this temperature the processes of dihydroxylation of the brucite layer Mg(OH)_2_ and the breaking of Si–O(Si) bonds in the silaxone (Si–O–Si) bridges of the hexagonal structure of the silicate layer of serpentinite apparently have not yet begun. The apparent beginning of the increase in Mg and Si in solution (Figure 11a), which indicates the beginning of processes leading to changes in the structure of the crystal lattice Mg_6_Si_4_O_10_(OH)_8_, is observed in a sample that has been heat-treated at 600 °C (No. 2). Significantly increased conversion of magnesium and silicon to solution due to the breakdown of the Mg_6_Si_4_O_10_(OH)_8_ crystal lettuce was found during the dissolution of samples (No. 3 and No. 4) thermoactivated at temperatures of 725 °C and 750 °C, respectively.

The active destruction of the crystal lattice structure and occurrence of the acid-base interactions involving MgO are also disclosed in terms of the pH change in the medium (Figure 11a, curve 3), the values of which increase in parallel with an increase in the Mg ^2+^ content in the solution, which additionally confirms the results of the X-ray phase analysis, during which the structural structure of the crystal lattice of serpentinite undergoes a substantial transformation. The dehydrocylation process results in periclase formation, as indicated by the increased intensity of periclase reflexes on x-rays of samples No. 3 and No. 4. Periclase is most likely the primary reason for the increased reactivity with acid, as its alkaline properties are higher than those of magnesium hydroxide, supporting a comparison of the value ∆G_ch.r._ and a thermodynamic evaluation of the reactions between MgO and Mg(OH)_2_ and sulfuric acid (Table 3).

Of the mineral phases discovered as separate in the product of thermolysis of the given serpentinite (up to 800 °C) (periclase, pyrop, almandin, forsterite, magnetite, diopside), in addition to periclase, there are insoluble or insoluble minerals in sulfuric acid. Magnesium oxide is more soluble in water and has a stronger alkaline property than the brucite-like components in Mg_6_Si_4_O_10_(OH)_8,_ which is also indicated by thermodynamic evaluation data for their sulfuric acid reactions (Table 3). The observed dissolution of serpentinite at different temperatures by the amount of magnesium, silicon and pH of the medium found in the solution is relatively similar up to a temperature of 600 °C (Figure 11a). Further, the marked characteristics (solution color, amount of Mg and Si, pH of the medium) for samples punctured in the temperature range of 600–750 °C undergo noticeable changes towards higher values.

In favor of the fact that the thermolysis of serpentinite in the temperature range of 600–750 °C occurs with the formation of the periclase phase, it can also be noted the previously discovered fact that when these samples interact with acid, higher increases in the temperature of the reaction medium are observed. The fact that with the increase in calcination temperature, the ratio of detected admixtures of metals such as iron, aluminum and calcium in the solution of Me_rates._^n+^/Me_ostat._^n+^ is evidently connected with the fact that in the process of dihydroxylation and its further decomposition, they also evidently participate in the formation of new crystals of pyrop, almandin, magnetite and diopside (Figure 11b–d).

The results of the observation of the nature of the solution of serpentinite sulfuric acid, which has been heat-treated at the above-mentioned temperatures, the X-ray phase studies of acid-insoluble residues, as well as the chemical analysis of the products of the solution thereof, clearly indicate that, during heat treatment of serpentinite in a temperature range of 600–750 °C, periclase (MgO) is formed in the composition of serpentinite as a result of the following changes, which significantly increase its alkaline properties and the ability to dissolve in acids.

## 4. Materials and Methods

Initially, in order to select the calcination temperatures, the thermal behavior of the serpentinite under investigation (Sp) was studied using differential thermal decomposition (DTA) and thermographic analysis (TGA) in the range of 200 °C to 1000 °C.

The heat treatment of serpentinite at selected temperatures based on the results of its DTA and TGA (600 °C, 725 °C and 750 °C) was as follows: 20 g of serpentinite was placed with a lid and treated in a muffle furnace, where the sample was heated at a set temperature (600 °C, 725 °C and 750 °C) for 1 h. After heat treatment, the weight of each sample was measured to calculate weight loss.

The acid treatment of the heat-treated sample was carried out with a sulfuric acid solution containing a stoichiometrically required amount (SRA) in relation to the quantity of magnesium content in the weighted charge (10 g) of serpentinite in a ratio of 1:1, and the volume of the solution was 100 mL, i.e., 100 mL of solution contains 1.0 SRA H_2_SO_4_ in relation to the quantity of magnesium in Mg (Sp).

The study of the interaction of heat-treated serpentinite samples with solutions was carried out as follows: 100 mL containing 1.0 SRA H_2_SO_4_ was placed in a glass (200 mL) and put in an aqueous thermostat, preheated to 80 °C. Into the heated solution, the heat-treated specimens individually prepared were introduced slowly in two or three moves during agitation, lasting 20 min, after which the suspension was transferred to the Büchner funnel with a vacuum water-jet pump. The insoluble sludge was filtered and washed 3 times with a little water. The filtered solution was poured into a glass (150 mL) and put in a thermostat for evaporation to about 100 mL, then transferred into a 100 mL flask, bringing the flask to the mark with water, and pH was measured. The concentration of residual acid after the interaction of serpentinite and acid was determined by titration of the filtrate with NaOH solution, at a concentration of 3.75 mol/L. The concentration of the residual acid was calculated using the formula: C_H2SO4_ = C_NaOH_·V_NaOH_/2V_filtrate._

To determine the mass of the dry residue, 50 mL of leachate was evaporated and dried at 105 °C to a constant weight. Acid-insoluble residue at 105 °C was similarly dried. The dry residue and sludge were then subjected to chemical analysis.

Chemical analyses were carried out on a JSM-6490LV device, JEOL (Tokyo, Japan), complete with the energy dispersion microanalyzer system INCAEnergy 350.

X-ray-phase spectra of heat-treated (TPS) and acid-treated (TPS) are obtained on the diffractometer D8Advance (Bruker, Billerica, MA, USA), Cu-K_α_, 40 kV tube voltage and 40 mA current. The processing of the resulting diffractograms and the calculation of the inter-plane distances were carried out using the EVA software (DIFFRAC.EVA V4). Sample decoding and phase finding were performed using the search/match program using the PDF-2 Powder Diffractometric Data Base (JCDD) and the information given in [17].

Thermal curves (TG, DTA) were recorded on “Q-DERIVATOGRAPH” (Paulik-Paulik-Erdey system derivative (MOM, Q-1500D, F. Paulik, J. Paulik, L. Erdey system, Budapest, Hungary).

Thermodynamic evaluation of probable reactions has been performed using the HSC Chemistry 6.0 program (A. Roine, HSC Chemistry, Metso:Outotec, Pori, 2021).

## 5. Conclusions

Thus, heat-treated serpentinite of the Zhitikarinsky deposit is formed in the narrow temperature range of 600–750 °C and, depending on the processing temperature, when exposed to acids, it exhibits various degrees of reactivity. The manifestation of increased active interaction of serpentinite with acid is most likely associated with the appearance of periclase (MgO) in its composition after their heat treatment in the temperature range 725–750 °C, as a result of thermal decomposition of the serpentinite structure (Mg_6_Si_4_O_10_(OH)_8_). Periclase has a higher alkaline property and solubility in water than the brucite-like components in the structural composition of the initial serpentinite. Higher heat treatment temperature (800 °C) serpentinite leads to a decrease in the reactivity of serpentinite in relation to acidic interactions.

## Figures and Tables

**Figure 1 molecules-29-03965-f001:**
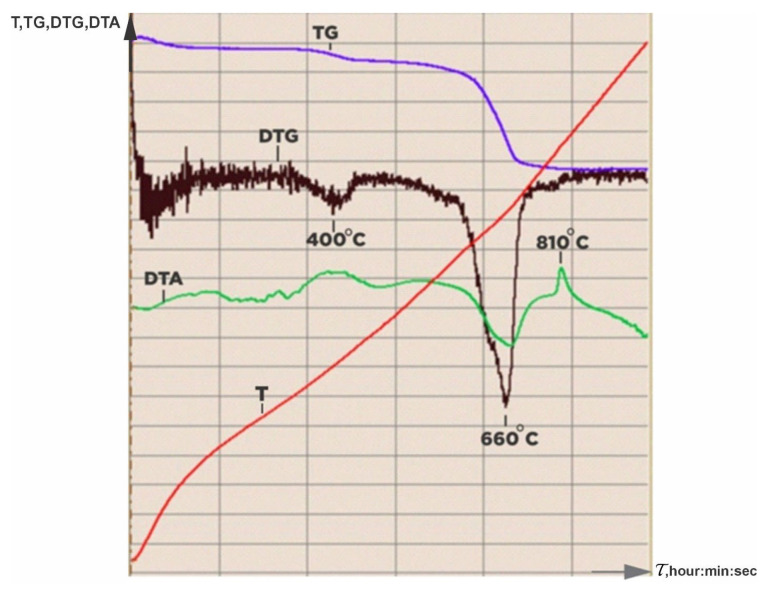
Derivatogram of the initial serpentinite, heat-treated at 105 °C.

**Figure 2 molecules-29-03965-f002:**
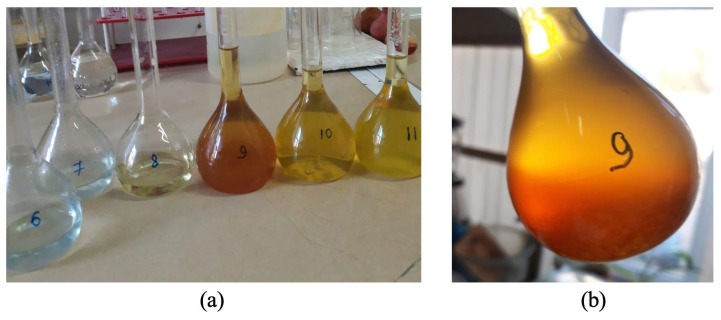
Change in color of the resulting solutions when serpentinite is dissolved in sulfuric acid depending on the heat treatment temperature of the initial serpentinite (**a**): 6—600 °C; 7—625 °C; 8—660 °C; 9—725 °C; 10—750 °C; 11—800 °C and (**b**): 725 °C—the next day.

**Figure 3 molecules-29-03965-f003:**
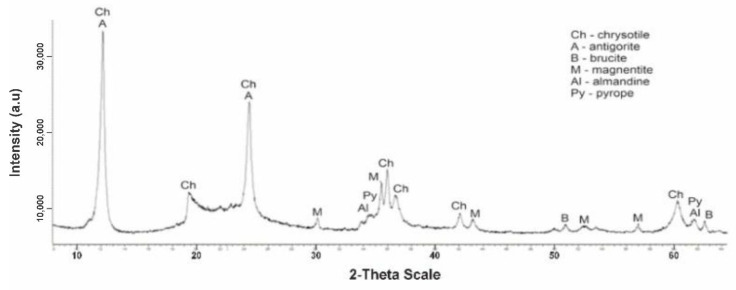
Diffractogram of the initial dry serpentine (sample No. 1, t = 105 °C).

**Figure 4 molecules-29-03965-f004:**
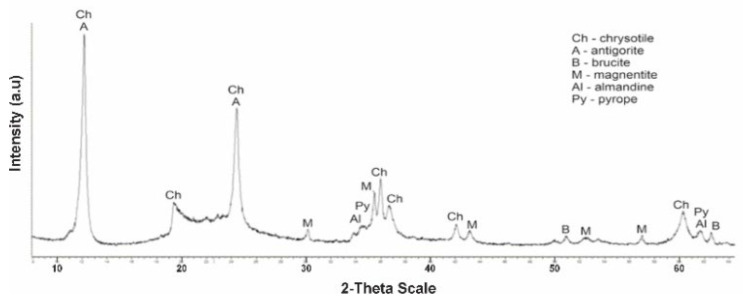
Diffractogram of the initial serpentinite (sample No. 1, 105 °C) after acid treatment (1.0 SRA H_2_SO_4_).

**Figure 5 molecules-29-03965-f005:**
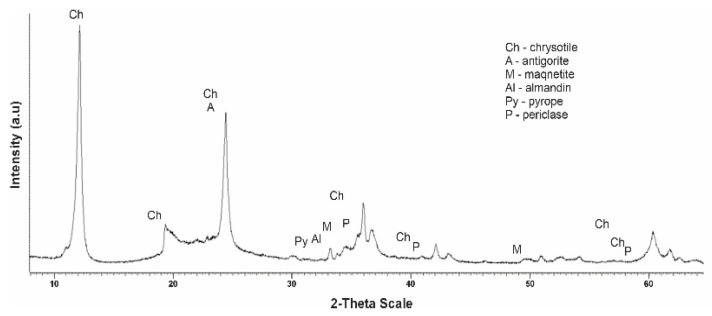
Diffractogram of serpentinite (sample No. 3, 725 °C) after acid treatment (1.0 SRA H_2_SO_4_).

**Figure 6 molecules-29-03965-f006:**
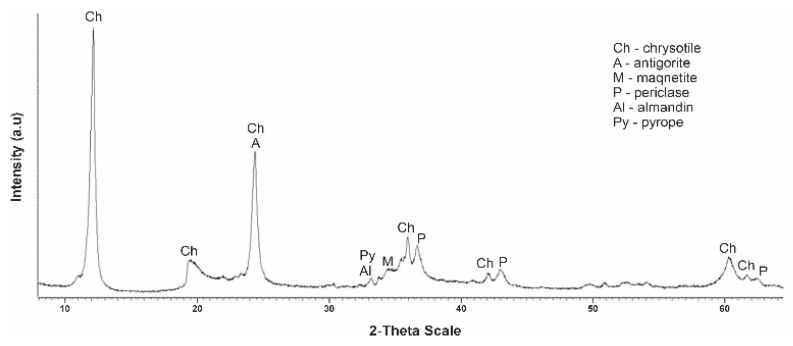
Diffractogram of serpentinite (sample No. 4, 750 °C) after acid treatment (1.0 SRA H_2_SO_4_).

**Figure 7 molecules-29-03965-f007:**
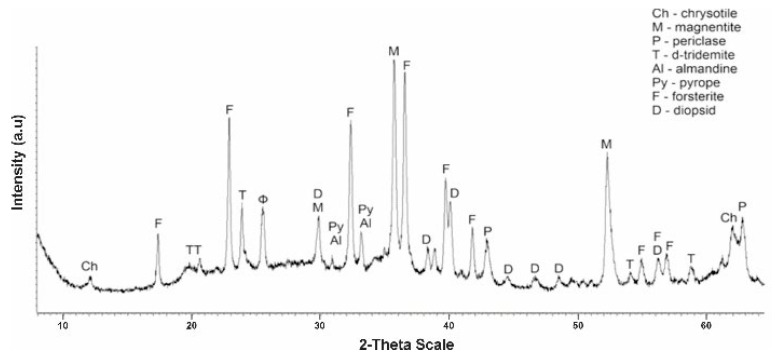
Tempered sample No. 3 serpentinite diffractogram at 725 °C (1 h).

**Figure 8 molecules-29-03965-f008:**
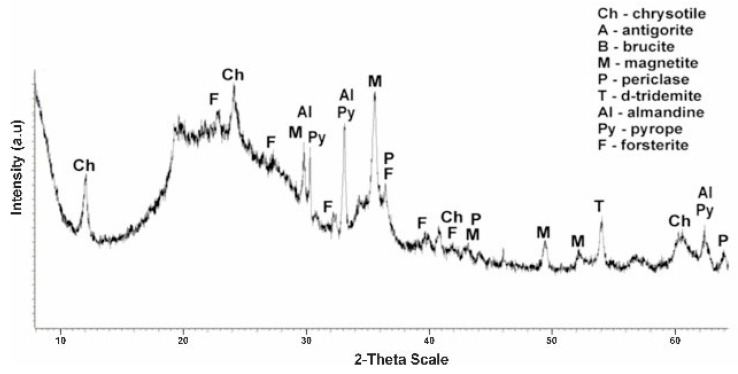
Diffractogram of No. 3 K serpentinite (725 °C, 1 h) obtained after acid treatment (1.0 SRA H_2_SO_4_).

**Figure 9 molecules-29-03965-f009:**
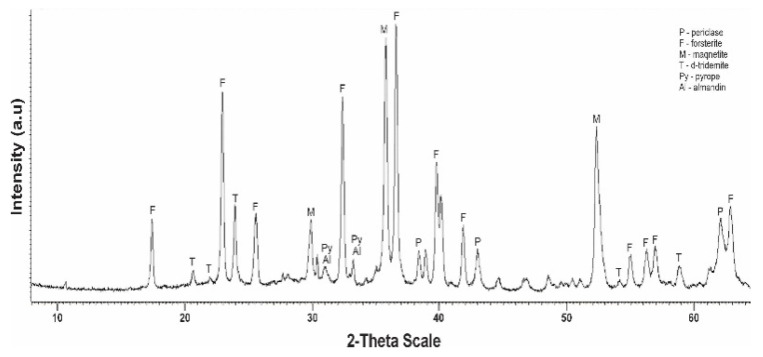
Diffractogram No. 4 of hardened (at 750 °C, 1 h) serpentinite.

**Figure 10 molecules-29-03965-f010:**
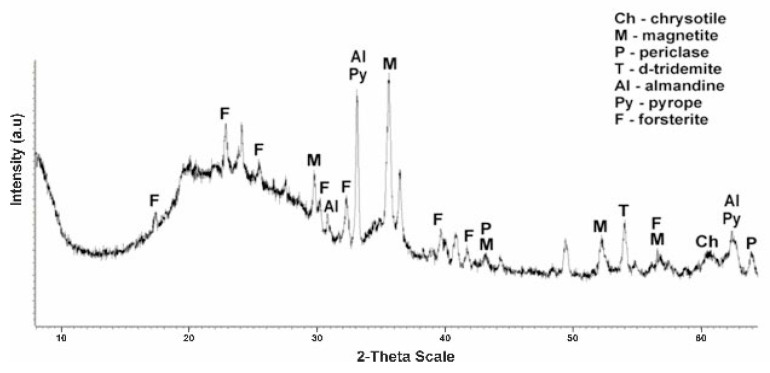
Diffractogram of No. 4 K serpentinite (750 °C, 1 h) obtained after acid treatment (1.0 SRA H_2_SO_4_).

**Figure 11 molecules-29-03965-f011:**
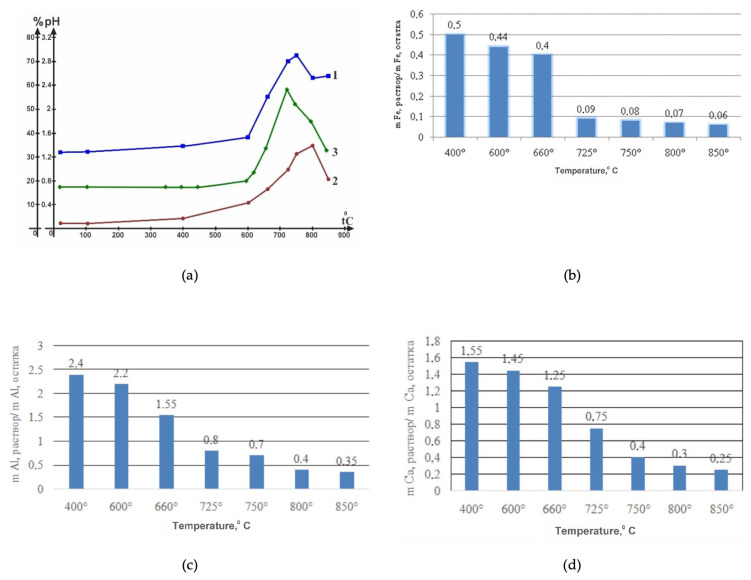
Output relationships (in %) (**a**) in magnesium sulfate solution (1), silicon (2), pH changes in medium (3) and distribution of iron (**b**), aluminum (**c**) and calcium (**d**) from serpentinite treatment temperature.

**Table 1 molecules-29-03965-t001:** Distribution of magnesium in the dissolution products depending on the processing temperature of the initial serpenitinite CH_2_SO_4_ = 1.0 KCC, t = 80 °C, V = const (100 mL).

No. Sample	Processing Temperature, t °C	The Amount of MgO in the Suspension, g	The Amount of MgO Transferred to the Solution		The Amount of MgO in the Insoluble Residue	
			g	%	g	%
1	105	4.16	1.95	47.0	2.18	52.2
2	600	4.53 *	3.32	73.2	1.19	26.1
3	725	4.59 *	3.86	84.0	0.69	15.0
4	750	4.66 *	4.08	87.5	0.57	12.2
5	800	4.68 *	3.80	82.0	0.81	17.3

Note: *—the amount of MgO in the suspension increases due to the loss of water during calcination.

**Table 2 molecules-29-03965-t002:** Thermodynamic evaluation of probable thermodecomposed reactions of serpentinite to periclase (MgO), temperature 725 °C.

Reaction	∆G_ch.r._, kJ/mol
Mg_3_Si_2_O_5_(OH)_4_ → MgO + 2MgSiO_3_ + 2H_2_O_(g)_	−129.812
Mg_3_Si_2_O_5_(OH)_4_ → 2MgO + MgSiO_3_ + 2H_2_O_(g)_	−91.023

**Table 3 molecules-29-03965-t003:** Thermodynamic evaluation of Mg(OH)_2_ and MgO sulfuric acid reactions.

Reaction	∆G_ch.r._, kJ/mol			
	Temperature, °C			
	40	60	80	100
Mg(OH)_2_ + H_2_SO_4_ = MgSO_4_ + 2H_2_O	−109.0	−109.0	−109.1	−109.2
MgO + H_2_SO_4_ = MgSO_4_ + H_2_O	−115.3	−115.2	−115.1	−115.0

## Data Availability

The data used to support the findings of this study are included within the article.

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
