# Peer review of "Thermoacid Behavior of Serpentinite of the Zhitikarinsky Deposit (Kazakhstan)"

_molecules, 2024, doi:10.3390/molecules29163965_

Round 1

Reviewer 1 Report

Comments and Suggestions for Authors

Regarding the work “Thermo and acid treatment of serpentinite from the Zhitikarinsky deposit (Kazakhstan)”.

• On page two the font must be uniform, including the font colour.

• It is suggested that the results and discussion be separated (page 2) and that they go after materials and methods (page 8)

• In Figure 1. Derivatogram of the initial serpentinite, heat-treated at 105°C. It is necessary to mark the symbology of the axes.

• In Figure 3. Diffractogram of the initial dry serpentine (sample No.1, t=105°C). This is an analysis by interplanar or Miller indices. Does not include the font. Nor is it mentioned whether the analytical method was used to calculate the phases or their percentage by calculating R. Figure 4. Diffractogram of the initial serpentinite (sample No.1, 105°C) after acid treatment (1.0 NAC H2SO4) is similar to Figure 3, they are incorrectly indexed since a specific peak or tetha angle shows more than one phase, which is incorrect, nor are the PDFs included, and the peaks with their interplanar distance or indexes are not indicated. Miller. In Figure 5. Diffractogram of serpentinite (sample No.3, 725°C) after acid treatment (1.0 NAC H2SO4), in a peak marking two phases, the theta angle is not clearly observed, but in the middle it overlaps Al and Py, not They are the same, almandine contains iron and pirope does not. It also happens in figure 6.

• The bibliography must be increased and updated.

Comments on the Quality of English Language

 Minor editing of English language required

Author Response

C1• On page two the font must be uniform, including the font colour.

R1: We agree. Corrected

C2• It is suggested that the results and discussion be separated (page 2) and that they go after materials and methods (page 8)

R2: We agree. Divided: “Results” and “Discussion”.

C3• In Figure 1. Derivatogram of the initial serpentinite, heat-treated at 105°C. It is necessary to mark the symbology of the axes.

R3: We agree. The symbolism of the axes was noted.

C4• In Figure 3. Diffractogram of the initial dry serpentine (sample No.1, t=105°C). This is an analysis by interplanar or Miller indices. Does not include the font. Nor is it mentioned whether the analytical method was used to calculate the phases or their percentage by calculating R. Figure 4. Diffractogram of the initial serpentinite (sample No.1, 105°C) after acid treatment (1.0 NAC H2SO4) is similar to Figure 3, they are incorrectly indexed since a specific peak or tetha angle shows more than one phase, which is incorrect, nor are the PDFs included, and the peaks with their interplanar distance or indexes are not indicated. Miller. In Figure 5. Diffractogram of serpentinite (sample No.3, 725°C) after acid treatment (1.0 NAC H2SO4), in a peak marking two phases, the theta angle is not clearly observed, but in the middle it overlaps Al and Py, not They are the same, almandine contains iron and pirope does not. It also happens in figure 6.

R4: All drawings based on “Diffractograms” have been reworked, formatted and indexed. The identification of MPR minerals was further clarified using data from [17]. The text of the descriptions of the diffraction patterns was corrected and supplemented taking into account the detected shortcomings.

C5• The bibliography must be increased and updated.

R5: We agree. The bibliography has been expanded and updated (up to 17 references).

C6: Minor editing of English language required

R: The quality of the English language may have been poor. Therefore, the text of the article has been completely re-edited.

PS. The authors of the article are grateful to Reviewer 1 for a detailed analysis of the material, correct comments and suggestions, which were taken into account in finalizing the materials of the article, which contributed to improving the quality of the article.

Reviewer 2 Report

Comments and Suggestions for Authors Remarks 1.“The thermal acid behavior of serpentinite...has been studied” In my opinion, "thermic and acid treatment behavior" is more appropriate in this case.   2. "...the formation of periclase (MgO) in the composition of serpentinite, after heat treatment in the temperature range of 725-750°C." Is it possible the formation of periclase (MgO) under such PT conditions?   3."The results were discussed using data obtained by conducting a thermodynamic assessment of probable reactions during the thermal acid treatment of serpentinite". Thermodynamic calculation methodology should be added.   4.“The phase compositions of heat-treated serpentinite were determined at 600-750°C, after their acid treatment in 1.0 M H2SO4”. It is necessary to justify why this temperature diapason was chosen.   5.“The purpose of this work is to determine the effect of thermal and acid treatment on the composition of the products of dissolution of the waste from the enrichment of chrysotile asbestos ore of the Zhitikarinsky deposit in sulfuric acid.” The purpose of the work must be justified. Why is there a need to treat the waste from the enrichment of chrysotile asbestos ore using sulfuric acid. Won't this result in even more dangerous waste? This should be clarified.   6. All obtained peaks in the presented XRD patterns (figs. 3-6) should be compared with the Jade library Joint committee on powder diffraction standards (JCPDS).   7.The presence of minerals such as pyrope, almandine, tridemite and periclase in acid-treated samples should be explained in the work.   8.The quality of the figures should be improved.    

Author Response

Remarks 

1.“The thermal acid behavior of serpentinite...has been studied” In my opinion, "thermic and acid treatment behavior" is more appropriate in this case. 

Response:  We agree. Topic changed

  1. "...the formation of periclase (MgO) in the composition of serpentinite, after heat treatment in the temperature range of 725-750°C." 

Is it possible the formation of periclase (MgO) under such PT conditions?  

Response: Yes. We understand that the issue is controversial. However, the discovered facts of the chemical behavior of serpentinite heat-treated at 600ºC, 725ºC and 750ºC, which are characterized by an increase in their reactivity expressed by an increase in the temperature of the environment from 88ºC to 95ºC (750ºC) (page 5), indicates the appearance of an active alkaline component in the system. Of the phases detected (by XRD), only periclase can exhibit such activity. In addition, from serpentinite heat-treated in the temperature range 600-750ºC, during their interactions with the acid solution, a relatively large amount of magnesium ions are extracted into the solution (Figure 7, a). Thermodynamic calculation ∆GÑ….Ñ€=f(T) of the thermal decomposition of serpentinite with the formation of periclase does not exclude the occurrence of such a reaction. Therefore, we included the results of calculating ∆GÑ….Ñ€=f(T) of such a reaction (additionally) in the article (see page 12, in the “Discussion” section). The presented experimental and calculated data indicate the possibility of the formation of periclase (MgO) under these conditions.

3."The results were discussed using data obtained by conducting a thermodynamic assessment of probable reactions during the thermal acid treatment of serpentinite". 

Thermodynamic calculation methodology should be added.   

Response: The thermodynamic calculation method is indicated on page 3 (in the “Research Methodology” section). The standard calculation program ∆GÑ….Ñ€=f(T) was used.

4.“The phase compositions of heat-treated serpentinite were determined at 600-750°C, after their acid treatment in 1.0 M H2SO4”. 

It is necessary to justify why this temperature diapason was chosen.  

Response: We agree. We proceeded from the purpose of this work. It is known that one of the main goals of studying serpentinites and waste from processing chrysotile raw materials is to use them as a source of magnesium. Heat treatment of serpentinite is one of the effective ways to increase their reactivity. The choice of the temperature range of 600-750ºÐ¡ is justified by the fact that in this range there is destruction (dehydroxylation) of the crystal lattice of the structural structure of serpentinite (Figure 1) and the formation of products (components) of its decomposition, which determine the nature of its dissolution in acids.

5.“The purpose of this work is to determine the effect of thermal and acid treatment on the composition of the products of dissolution of the waste from the enrichment of chrysotile asbestos ore of the Zhitikarinsky deposit in sulfuric acid.” 

The purpose of the work must be justified. Why is there a need to treat the waste from the enrichment of chrysotile asbestos ore using sulfuric acid. Won't this result in even more dangerous waste? This should be clarified.   

Response: We agree. It is possible that the purpose of the work was not formulated quite adequately and clearly, so the purpose of the work was revised and adjusted.

The problems that arise when processing serpentinites using acid methods have not yet been resolved. They have both technological and economic characters. Proof of this situation is that despite the large number and versatility of research being carried out in the world, there is still no method of processing it acceptable for practical use. The choice of sulfuric acid for processing is justified by the fact that: 1) The most accessible and cheap; 2) Research (they are known from the literature) has established that among mineral acids (HCl, HNO3 and H2SO4), sulfuric acid is better at revealing magnesium silicates with a serpentinite structure.

The choice of the temperature range of 600-750ºÐ¡ is also justified by the fact that in the temperature range the chrysotile structure of asbestos is destroyed (Figure 1), which is associated with its harmfulness and danger. No hazardous components were found among thermal acid products processed at 600-750ºC, that is, no hazardous waste is generated.

  1. All obtained peaks in the presented XRD patterns (figs. 3-6) should be compared with the Jade library Joint committee on powder diffraction standards (JCPDS). 

Response: Processing of the obtained diffraction patterns and calculation of interplanar distances were carried out using EVA software. Sample interpretation and phase search were carried out using the Search/match program using the PDF-2 Powder Diffractometric Database (JCDD) and the information given in [17]. Due to lack of time to respond (until July 12, 2024), we were not able to compare with the standards of the Jade Joint Committee on Powder Diffraction (JCPDS) library, but we understand that this must be done. Thank you!

7.The presence of minerals such as pyrope, almandine, tridemite and periclase in acid-treated samples should be explained in the work. 

Response: We agree. Yes. These minerals (pyrope, almandine, tridemite and periclase) were found in samples after acid treatment. Pyrope and almandine were present in the original samples (Figure 3), apparently they do not undergo much change during thermal acid treatment; tridemite (SiO2) is usually formed and one of the products during acid treatment of serpentinites from silicic acids (after drying acid-insoluble residues at 105ºC). Periclase, according to our version, is a product of the thermal decomposition of Mg6Si4O10(OH)8 at 600-750ºÐ¡, apparently during acid treatment under these conditions its complete dissolution does not occur, possibly related to the crystallization temperature of MgO (the hydraulic activity of which decreases with increasing crystallization temperature), but we don't have a detailed explanation yet.

8.The quality of the figures should be improved.

Response: We agree. The quality of the drawings has been improved.

PS. The authors of the article are grateful to Reviewer 2 for a detailed analysis of the material, correct comments and suggestions, which were taken into account in finalizing the materials of the article. We think that the revised article, taking into account the Reviewer’s comments, turned out to be more structured and meaningful. Thank you!

Round 2

Reviewer 2 Report

Comments and Suggestions for Authors

The revised manuscript can be published, in my opinion.